# {Gd^III^_7_} and {Gd^III^_14_} Cluster Formation Based on a Rhodamine 6G Ligand with a Magnetocaloric Effect

**DOI:** 10.3390/molecules29020389

**Published:** 2024-01-12

**Authors:** Lin Miao, Cai-Ming Liu, Hui-Zhong Kou

**Affiliations:** 1Engineering Research Center of Advanced Rare Earth Materials (Ministry of Education), Department of Chemistry, Tsinghua University, Beijing 100084, China; 2Beijing National Laboratory for Molecular Sciences, Center for Molecular Science, Institute of Chemistry, Chinese Academy of Sciences, Beijing 100190, China; cmliu@iccas.ac.cn

**Keywords:** lanthanide oxygen cluster, magnetocaloric effect, magnetic refrigerant materials, rhodamine ligand

## Abstract

Heptanuclear {Gd^III^_7_} (complex **1**) and tetradecanuclear {Gd^III^_14_} (complex **2**) were synthesized using the rhodamine 6G ligand HL (rhodamine 6G salicylaldehyde hydrazone) and characterized. Complex **1** has a rare disc-shaped structure, where the central Gd ion is connected to the six peripheral Gd^III^ ions via CH_3_O^−^/μ_3_-OH^−^ bridges. Complex **2** has an unexpected three-layer double sandwich structure with a rare μ_6_-O^2−^ ion in the center of the cluster. Magnetic studies revealed that complex **1** exhibits a magnetic entropy change of 17.4 J kg^−1^ K^−1^ at 3 K and 5 T. On the other hand, complex **2** shows a higher magnetic entropy change of 22.3 J kg^−1^ K^−1^ at 2 K and 5 T.

## 1. Introduction

The exploration of new compounds in the field of coordination chemistry reveals many fascinating structures and properties. Among them, rare earth ions play an important role in the assembly of polynuclear clusters [1,2,3]. Due to their large ion radius and inherent ability to participate in various coordination environments, rare earth ions can promote the formation of polymetallic clusters [4,5,6,7,8,9,10,11,12,13,14,15,16,17,18,19,20,21,22]. Multinuclear rare earth clusters with different structural types, such as {Ln_18_} [23], {Ln_28_} [24], {Ln_34_} [25], {Ln_36_} [26], {Ln_37_} [27], {Ln_48_} [28], {Ln_60_} [29], {Ln_72_} [30], {Ln_104_} [31], {Ln_140_} [32], etc., were synthesized, and the largest even has a {Nd_288_} structure [33]. These polynuclear clusters usually exhibit fascinating properties, such as single-molecule magnetism [8,12,14,15,19,30], luminescence [23,24], magneto-optical properties [34] and proton-conductive properties [35].

The formation of lanthanide hydroxide clusters involves the hydrolysis of lanthanide ions in the presence of ligands. The hydrolysis process leads to the formation of small lanthanide hydroxide units, which then assemble to form larger clusters. The size and structure of the clusters can be controlled by adjusting the hydrolysis conditions and the choice of ligands. The hydrolysis-induced assembly mechanism of lanthanide hydroxide clusters is still not fully understood due to the elusive coordination configurations of lanthanide ions and the limited characterization methods available. However, recent studies have made progress in determining the intermediate species and the pathways of cluster formation [22]. The formation of high-nuclearity lanthanide clusters is believed to involve the assembly of low-nuclearity subunits, which are formed according to initial hydrolysis. These small units are then connected to form the final cluster structure. Understanding the formation mechanism of lanthanide hydroxide clusters is important for the development of new functional materials and their applications in various fields. In addition, the interactions between the rare earth centers can affect the physical properties of the clusters. The ligand framework around these ions also plays an important role in determining the geometry, stability and functionality of the obtained clusters. By adjusting the ligand, scientists can strategically guide the self-assembly process, thus generating the required multi-core architecture.

Certain gadolinium compounds have shown excellent magnetocaloric properties [36,37,38], such as Gd(HCOO)_3_ [39], Gd(OH)_3_ [40], Gd_2_O(OH)_4_(H_2_O)_2_ [40], GdPO_4_ [41], Gd(OH)CO_3_ [42], GdF_3_ [43], Gd(OH)F_2_ [44] and Ba_2_Gd(BO_3_)_2_X (X = F, Cl) [45]. The magnetocaloric effect (MCE) in Gd is particularly strong, making it an ideal material for use in extremely low-temperature magnetic refrigeration systems. Magnetic refrigeration is a technology that uses a magnetic field to cool objects. Magnetic refrigeration materials play an increasingly important role in the future of social development, particularly in the area of energy efficiency and sustainability. In addition, molecular clusters with diverse structures also exhibit enormous magnetic cooling potential and exhibit many physical properties, such as chirality, spin crossover, fluorescence, etc.

Rhodamine-derived ligands show ring-opened or ring-closed structures and can form rare earth [46,47,48,49] or transition metal [50,51] complexes. They are mostly mononuclear or low-nuclearity complexes. We are interested in forming high-nuclearity clusters based on rhodamine ligands using the hydrolysis approach. In this work, we used the rhodamine 6G salicylaldehyde hydrazone ligand (Figure 1) to synthesize two different metal complexes, hexagonal heptanuclear {Gd^III^_7_} [Gd_7_(L)_6_(μ_2_-CH_3_O)_4_(μ_3_-CH_3_O)_4_(μ_3_-OH)_4_(NO_3_)_2_]NO_3_·10CH_3_CN·10CH_3_OH·2H_2_O (complex **1**) and tetradecanuclear {Gd^III^_14_} Gd_14_(H_0.5_L)_8_(μ_6_-O)(μ_4_-O)_2_(μ_3_-OH)_16_(NO_3_)_16_·9.5CH_3_CN·2CH_3_OH·11H_2_O (complex **2**), with an unexpected three-layer double sandwich structure. These two complexes have an excellent magnetic refrigeration performance, and their magnetic entropy changes are 17.4 J kg^−1^ K^−1^ at 3 K and 5 T and 22.3 J kg^−1^ K^−1^ at 2 K and 5 T, respectively.

## 2. Results and Discussion

### 2.1. Synthesis and Characterization

Rhodamine 6G-type ligands have attracted significant attention in the realm of fluorescent sensors. Previous studies have primarily focused on the synthesis of mononuclear rare earth or transition metal complexes [46,47,48,49,50,51,52]. However, these ligands have yet to be fully unearthed. We speculate that the reaction between Ln^III^ and the rhodamine 6G ligands might lead to high-nuclearity lanthanide complexes under high pH values. Using the hydrolysis strategy, we have successfully synthesized two new Gd^III^ clusters, **1** and **2** (Figure 1). Additionally, our research has unveiled the influential factors that impact the synthesis process, including the ratio of central ions to ligands, solvents, alkalines and reaction temperature.

The synthetic procedure for the two clusters is similar, with the exception of the molar ratio of Gd:L, i.e., an excess amount of gadolinium nitrate was used in the synthesis of {Gd_14_}. Both complexes were prepared using the reaction of the ligand HL with gadolinium nitrate in a mixed solution of methanol and acetonitrile. A quantity of triethylamine was used to induce the ring closure of the HL and hydrolysis, which was characterized using the UV-Vis spectra of the reactant mixtures (Figure 2). The absorption at 535 nm due to the ring-opened xanthene disappeared after the addition of four times the amount of Et_3_N, revealing that the HL ligands were ring-closed [46,47,48]. The peak at 400 nm is most likely due to the conjugated salicylaldehyde hydrazone group, corresponding to the yellow color of the final products. The resulting mixture was left undisturbed at room temperature for one week, facilitating the formation of yellow plate-like single crystals for {Gd_7_}. The reactant mixture was heated at 60 °C in an oven for three days, resulting in the formation of yellow cubic-shaped samples of {Gd_14_}. These two complexes tend to lose solvents in the air. When the crystals are taken out of the solution, they turn from yellow to red at room temperature, and they lose their crystallinity. The powder XRD pattern for Gd_7_ illustrates that the main strong peaks show disagreement with those simulated, indicating the desolvation of the crystals (Appendix A). Thermogravimetric analysis (TGA) shows that the crystals lose all their crystallization solvent molecules to give a desolvated solid (Appendix A). The peaks at low 2θ angles in the PXRD data for Gd_14_ are approximately consistent with those simulated, and the TGA data show a weight loss of 8.5% in the temperature range 41–112 °C, which is in agreement with the calculated data of 7.7%. The infrared spectra of complexes **1** and **2** (Appendix A) show peaks at 1363 cm^−1^ and 1380 cm^−1^, respectively, as is characteristic of nitrate anions. The disappearance of the peak at 1680 cm^−1^ for HL illustrates the coordination of aryl oxygen toward Gd^III^. The strong peak at 1620 cm^−1^ is most likely due to the C=N stretching vibration for the Schiff base ligand L^−^. The UV-Vis and fluorescent spectra (Figure 2) for the reactant solutions of HL and Gd(NO_3_)_3_ with different amounts of Et_3_N were measured in the methanolic solution ([HL] = [Gd^3+^] = 10 μM, λ_ex_ = 354 nm). The colorless ligand shows no absorption in the visible region; however, after the addition of Gd^III^, the ligand becomes ring-opened owing to the coordination and has a strong typical absorption at 533 nm. With the addition of Et_3_N, the absorption gradually decreases owing to the ring-closure of HL in alkaline conditions. The free ligand HL in MeOH has an emission at 555 nm, which is due to the presence of small quantities of the ring-opened ligand. After the addition of Gd^III^, the ligand HL becomes ring-opened, and emits yellow light with a wavelength of 580 nm. When three times the amount of Et_3_N was added, the emission nearly disappeared, indicating that almost all the HL ligands were in the ring-closed form. The complexes are slightly soluble in MeOH, and the solutions (10 μM) show an intense absorption at 398 nm, corresponding to the yellow color of the complex with ring-closed L^−^ ligands (Appendix A). The emissions at 546 nm for Gd_7_ and 550 nm for Gd_14_ (λ_ex_ = 354 nm) potentially come from the ring-opened xanthene in the complexes, and although the ring-opened dissociation is scant, the emission is strong. The weak emission at 484 for Gd_7_ and 489 nm for Gd_14_ is most likely due to the Schiff base part of the ligand L^−^ (Appendix A).

### 2.2. Structure

A yellow single crystal of complex **1** was selected for single crystal X-ray diffraction at 100 K. Complex **1** crystallizes in the monoclinic system with the *P*2_1_/n space group. The volume of the unit cell is very large with a monoclinic system, and therefore the diffraction data are not optimal. The command MASK was used during the structural refinement, and 452 electrons were masked per formula unit, which account for the missing NO_3_^−^ anion and 10 acetonitrile, 10 methanol and 2 H_2_O molecules, with their total electrons being 451. The crystallographic data and selected bond distances and angles are given in Appendix A. The asymmetric unit cell contains three crystallographically independent {Gd^III^_7_} moieties. Because they have similar molecular structures, only one of them is described in detail as a representative. The coordination number of each Gd^III^ ion in {Gd_7_} (Figure 3) is between 7 and 9. Their coordination patterns are shown in Appendix A. Using the SHAPE software (version 2.1) for calculation, their coordination patterns were obtained, as shown in Appendix A. The central Gd1 ions are coordinated by eight oxygen atoms and have a square antiprism structure *D*_4d_. Among the eight oxygen atoms, four are μ_3_-OH^−^ and another four are μ_3_-CH_3_O^−^. The Gd–O_hydroxy_ bond distances are in the range of 2.316(9)–2.358(9) Å, slightly shorter than that of the Gd–O_methoxy_ bonds (2.362(9)–2.404(6) Å). The remaining six Gd^III^ ions are evenly distributed around the central Gd^III^ ion, forming a saddle-shaped structure, which is relatively rare in rare earth complexes [53] (Appendix A). Each peripheral Gd^III^ ion is chelated by one ring-closed ligand L^−^, and is bridged to the adjacent peripheral Gd^III^ ions by μ_2_-CH_3_O^−^ or phenoxy oxygen atoms (Figure 4). The Gd–O_methoxy_ bond distances are in the range of 2.261(10)–2.298(10) Å, comparable to that of the Gd–O_phenoxy_ bonds (2.189(10)–2.433(9) Å). The central Gd^III^ ion is connected to the six peripheral Gd^III^ ions via the μ_3_-OH^−^ and μ_3_-CH_3_O^−^ bridges, giving rise to a Gd_7_ core, as shown in Figure 3b. The bridging Gd–O–Gd bond angles are in the range of 92.9(3)–110.6(4)° for μ_3_-OH^−^, 92.9(3)–99.2(3)° for μ_3_-CH_3_O^−^, 110.4(4)–112.8(4)° for μ_2_-CH_3_O^−^ and 98.2(3)–98.4(3)° for μ_2_-O_phenoxy_, respectively, with intramolecular Gd–Gd separations of 3.5064(11)–3.808(11) Å. The neighboring Gd_7_ molecules are connected via weak intermolecular forces and no obvious intermolecular hydrogen bonds are found based on the squeezed model of the crystal data.

As for complex **2**, a yellow cube-shaped single crystal was selected for single crystal X-ray diffraction at 173 K. A solvent mask was used, and 183.4 electrons were found at a volume of 3448.9 Å^3^ in nine voids per unit cell, which is consistent with the presence of 2 CH_3_OH, 2 H_2_O and 1.5 CH_3_CN molecules per formula unit with 178 electrons. The crystallographic data and selected bond distances and angles are given in Appendix A. Complex **2** crystallizes in the tetragonal system with a space group of *P*4/n and has a *D*_4h_ symmetry. The asymmetric unit contains 1/4 of the tetradecanuclear molecule and there are five different kinds of nine-coordinated Gd^III^ ions (Gd1-Gd5, Figure 5b). They have three different coordination modes (Appendix A) and their coordination patterns are shown in Appendix A. The tetradecanuclear cluster core is neutral and has a highly symmetrical three-layer double sandwich structure (Figure 5b). In the structure, four nine-coordinated Gd^III^ ions form a square plane layer, and a nine-coordinated Gd^III^ ion is located between the layers. The distance between the two layers is 5.792 Å. The center of the middle layer is six-coordinated μ_6_-O^2−^, while the outer layers on both sides are μ_4_-O^2−^. The sandwiched Gd^III^ ions and the square-shaped layer are connected by μ_3_-OH^−^ ions. The Gd ions are linked together by hydroxo bridges, forming a [Gd_14_(μ_6_-O)(μ_4_-O)_2_(μ_3_-OH)_16_] core. This core contains one octahedral [Gd_6_(μ_6_-O)(μ_3_-OH)_8_] unit that shares two apexes with two [Gd_5_(μ_4_-O)(μ_3_-OH)_4_] square pyramid moieties. The cluster core is surrounded by eight ring-closed L^−^ ligands. Additionally, the Gd^III^ of the middle layer is coordinated with two nitrate ions, and the Gd^III^ of outer layers on both sides is coordinated with a nitrate ion and a tridentate ring-closed L^−^ ligand. The square plane layers are bridged by phenolic oxygen on the ligand, as shown in Figure 4b. Unlike in complex **1**, all the ligands L^−^ are involved in the bridging, and none of the coordination modes shown in Figure 4a are present in complex **2**.

As for the coordination environment of the Gd_14_ ions, three different types can be divided into Gd1 and Gd5 and their equivalences; Gd2 and Gd4; Gd3 and its equivalences. The Gd1 or Gd5 ion is surrounded by two μ_3_-OH^−^, two μ_2_-phenoxy, one μ_4_-O^2−^ and one L^−^ ligand, with Gd–O/N bond distances of 2.320(7)–2.587(19) Å for Gd1 and 2.292(8)–2.601(11) Å for Gd5. The Gd2 or Gd4 ion is coordinated by eight μ_3_-OH^−^ and one μ_6_-O^2−^, with Gd–O bond distances of 2.434(9)–2.617(13) Å for Gd2 and 2.454(8)–2.716(13) Å for Gd4. The Gd3 ion is coordinated by one μ_6_-O^2−^, four μ_3_-OH^−^ and two nitrate anions, with Gd–O bond distances of 2.335(8)–2.555(9) Å. The rare μ_6_-O^2−^-bridged Gd_6_ core has an elongated octahedral symmetry, with equatorial Gd–O bond distances of 2.4822(8) Å and axial Gd–O bond distances of 2.617(13)/2.716(13) Å. The intermetallic Gd–Gd separations within the Gd_14_ core are similar and are in the range of 3.5099(11)–3.882(1) Å. The Gd–O–Gd bond angles via the μ_3_-OH^−^ bridges are in the range of 95.9(3)–107.8(4)°, and those via the phenoxy bridges are similar (97.5(3) and 97.8(3)°). The acetonitrile molecules are hydrogen-bonded to the HN groups of L^−^. No intermolecular π–π contact is observed. The nearest intermolecular Gd–Gd separation is 13.73 Å.

Although there are many reports on tetranuclear clusters of planar quadrilateral [54,55] and nona-nuclear molecules of the double-layer sandwich type [56,57], a rare earth molecular structure in the form of a three-layer double sandwich is uncommon. Similar molecules have been reported before, as shown in Appendix A. Tetradecanuclear hydroxo–lanthanide acetylacetonato complexes, formulated as Ln_14_(μ_4_-OH)_2_(μ_3_-OH)_16_(μ-η^2^-acac)_8_(η^2^-acac)_16_ (Ln = Tb and Eu, acac^−^ = acetylacetonato) [58], and chiral tetradecanuclear hydroxo-lanthanide clusters, as Ln_14_(μ_4_-OH)_2_(μ_3_-OH)_16_(μ-η^2^-acac)_8_(η^2^-acac)_16_·6H_2_O (Ln = Dy and Tb) [59], have been reported. The ligands used in these two works are both based on acetylacetonato, but the present tetradecanuclear {Gd^III^_14_} is a completely different ligand, i.e., ring-closed rhodamine L^−^. The use of ortho-nitrophenolate exhibited the tetradecanuclear H_18_[Ln_14_(μ-η^2^-o-O_2_N-C_6_H_4_O)_8_(η^2^-o-O_2_N-C_6_H_4_O)_16_(μ_4_-O)_2_(μ_3_-O)_16_] (Ln = Dy and Tm; o-O_2_N-C_6_H_4_O^−^ = o-nitrophenolate) [60]. Despite the above similar Ln_14_ complexes, the μ_6_-O^2−^ in **2** is unique among them.

It is worth mentioning that similar hexadecanuclear molecules [Eu^III^_16_(tfac)_20_(CH_3_OH)_8_(μ_3_-OH)_24_(μ_6_-O)_2_] have also been reported based on trifluoroacetylacetone (tfac^−^) [22]. In addition to the tetradecanuclear {Eu_14_} core, there are another two Eu ions on the opposite sides (Appendix A). This study provides insights into the formation, evolution and assembly of lanthanide hydroxide clusters. The formation of {Gd_7_} and {Gd_14_} in this work further verifies that hydrolysis under high pH values is an effective way of constructing high-nuclearity Ln^III^ species.

### 2.3. Magnetic Measurements

The temperature dependence of the magnetic susceptibility of complexes **1** and **2** is measured under a 1000 Oe magnetic field in the range of 2–300 K (Figure 6). At room temperature, the *χ*_M_*T* values of 54.1 cm^3^ K mol^−1^ for {Gd_7_} and 109.0 cm^3^ K mol^−1^ for {Gd_14_} are close to the theoretical values of 55.09 cm^3^ K mol^−1^ for heptanuclear and 110.18 cm^3^ K mol^−1^ for tetradecanuclear uncoupled Gd^III^ (*S* = 7/2, *g* = 2, *C* = 7.87 cm^3^ K mol^−1^ per Gd), respectively. For **1**, upon lowering the temperature, the *χ*_M_*T* value slightly decreases to 49.76 cm^3^ K mol^−1^ at 20 K and then rapidly falls to 30.22 cm^3^ K mol^−1^ at 2 K. Complex **2** exhibits a similar behavior: when lowering the temperature, the *χ*_M_*T* value slightly decreases to 95.4 cm^3^ K mol^−1^ at 30 K, and then rapidly falls to 33.0 cm^3^ K mol^−1^ at 2 K. These changes indicate the presence of dominant antiferromagnetic interactions between the Gd^III^ ions in the clusters. The data can be perfectly fitted to the Curie–Weiss law, giving *C* = 54.44 cm^3^ K mol^−1^ and *θ* = −1.71 K for {Gd_7_} and *C* = 110.50 cm^3^ K mol^−1^ and *θ* = −4.63 K for {Gd_14_} (Appendix A). The larger absolute *θ* value in {Gd_14_} suggests that the antiferromagnetic interaction is stronger than that for {Gd_7_}.

The field dependence of the magnetizations (*M*) for complexes **1** and **2** was measured in the temperature range of 2–10 K (Appendix A). It can be seen that the magnetization has not reached saturation at 5 T and 2 K. At 2 K, the experimental maximum magnetization values of 47.03 Nβ for **1** and 89.35 Nβ for **2** are lower than the theoretical saturation value of Gd^III^ (49 Nβ for **1** and 98 Nβ for **2**, respectively), which may owe to the antiferromagnetic coupling, and a higher magnetic field is needed to suppress the magnetic coupling effect. For complexes **1** and **2**, the experimental *M–H* curves at 2–10 K lie below the calculated Brillouin curve for non-interacting *S*_Gd_ spins (Appendix A), also suggesting the presence of overall intermetallic antiferromagnetic coupling. The difference between the experimental and calculated values for Gd_14_ is obviously larger than that for Gd_7_, indicating that the former shows stronger antiferromagnetic coupling than that of the latter. The presence of μ_6_-O^2−^-bridged Gd_6_O moiety may be responsible for this.

The half-filled 4f electronic configuration in Gd^III^ ions makes them magnetically isotropic. This makes gadolinium a valuable material in various applications, especially magnetic refrigeration. Thus, the magnetocaloric effect (MCE) of complexes **1** and **2** was studied using the Maxwell equation:$$\Delta S_{m}{\left(T\right)}_{\Delta H}=\int {\left[\frac{\partial M\left(T,H\right)}{\partial T}\right]}_{H}dH$$

At 3 K and ∆*H* = 5 T, the value of −∆*S*_m_ is 17.44 J kg^−1^ K^−1^ for **1** (Figure 7a), which is slightly lower than the expected value of 14.56*R* (25.25 J kg^−1^ K^−1^) calculated for seven uncorrelated Gd^III^ ions using the equation −∆*S*_m_ = n*R*ln(2*S* + 1) (*R*≈8.314 J mol^−1^ K^−1^). The value of −∆*S*_m_ for **2** is 22.30 J kg^−1^ K^−1^ at 2 K and ∆*H* = 5 T (Figure 7b), which is close to the expected value of 29.11*R* (28.72 J kg^−1^ K^−1^) calculated for 14 uncorrelated Gd^III^ ions. To improve the magnetic refrigeration effect of the gadolinium clusters [37], several approaches can be considered. Firstly, the experimental conditions can be optimized. For instance, the temperature and magnetic field can be carefully controlled to ensure the most efficient operation of the gadolinium clusters. The thermal conductivity of the environment and the pressure during the refrigeration cycle can also be adjusted to minimize the energy loss. Secondly, the chemical composition of the clusters can be varied. Gadolinium can be alloyed with other metals to create compounds with different magnetic properties. Thirdly, the size of the clusters can be optimized. The optimal size will depend on the specific setup and application, but generally smaller clusters have a higher surface-to-volume ratio, which leads to more efficient heat exchange and therefore a stronger refrigeration effect. However, too-small clusters may also suffer from higher energy barriers between spin states, which can decrease the magnetic entropy change. Overall, a combination of these strategies can be used to improve the magnetic refrigeration effect of gadolinium clusters for various applications, such as cryogenic cooling of scientific instruments, temperature control in electronics and energy-efficient refrigeration in households and industries.

## 3. Materials and Methods

### 3.1. Synthesis and Preparations

All of the reagents we used were commercially available and used without further purification. The ligand HL we used was synthesized according to the method in the literature [46,47,48,49].

#### 3.1.1. Synthesis of the Ligand HL

Step 1: Synthesis of Rhodamine 6G hydrazide. Solid rhodamine 6G (10 mmol, 4.5 g) was dissolved in ethanol (100 mL). During stirring, 80% hydrazine hydrate solution (10 mL) was slowly added, and then the mixture was heated at 80 °C. After refluxing for 3 h, the reaction solution was cooled to room temperature. A light-colored precipitate was generated from the orange solution, which was collected using filtration, and washed with deionized water and ethanol until the color of the product turned white. Yield: about 4 g.

Caution: As hydrazine hydrate is a highly toxic reagent, the operations should be performed in a fume hood with care.



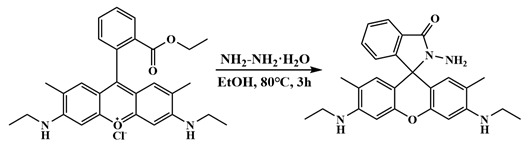



Step 2: Synthesis of HL. Rhodamine 6G hydrazide (5 mmol, 2.22 g) was suspended in 100 mL ethanol under stirring and heating at 80 °C, to which salicylaldehyde (10 mmol, 1 mL) was slowly added. The reaction continued at 80 °C for 3 h, and then the mixture was cooled to room temperature to give a light-colored solid. Filtering and washing with deionized water and ethanol generated a white solid powder of HL (about 2 g). The solid was used without further purification.



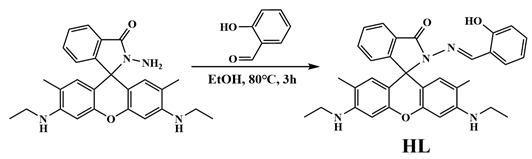



#### 3.1.2. Synthesis of [Gd_7_(L)_6_(μ_2_-CH_3_O)_4_(μ_3_-CH_3_O)_4_(μ_3_-OH)_4_(NO_3_)_2_]NO_3_·10CH_3_CN·10CH_3_OH·2H_2_O (**1**)

The ligand HL (0.2 mmol, 106 mg) was suspended in a mixed solution of MeOH (10 mL) and MeCN (10 mL), to which Gd(NO_3_)_3_·6H_2_O (0.25 mmol, 108 mg) was added, generating a red solution. The solution was heated and stirred for a few minutes and then triethylamine (300 µL, 2.16 mmol) was added to give a yellow solution. The mixture was filtered and the filtrate was placed undisturbed at room temperature. Yellow crystals suitable for X-ray diffraction analysis were collected after about 7 days. Yield: about 20%. Desolvated samples were used for the elemental analysis and physical measurements. Elemental analysis calcd. (%) for C_206_H_214_N_27_O_39_Gd_7_ (4793.23): C, 51.62; H, 4.50; N, 7.89. Found: C, 51.79; H, 4.91; N, 7.92.

#### 3.1.3. Synthesis of Gd_14_(H_0.5_L)_8_(μ_6_-O)(μ_4_-O)_2_(μ_3_-OH)_16_(NO_3_)_16_·9.5CH_3_CN·2CH_3_OH·11H_2_O (**2**)

The ligand HL (0.2 mmol, 106 mg) was suspended in the mixed solution of MeOH (10 mL) and MeCN (10 mL). Gd(NO_3_)_3_·6H_2_O (0.35 mmol, 150 mg) was added to generate a red solution. After heating and stirring for a few minutes, triethylamine (150 µL, 1.08 mmol) was added, giving rise to a yellow solution. The solution was filtered and the filtrate was heated in an oven at 60 °C. Yellow crystals suitable for X-ray diffraction analysis were obtained after about 2 days and collected. Yield: about 30%. Elemental analysis calcd. (%) for C_285_H_330.5_N_57.5_O_104_Gd_14_ (8427.05): C, 39.62; H, 3.95; N, 9.56. Found: C, 39.27; H, 4.10; N, 9.25.

### 3.2. Physical Measurements

The elemental analyses (C, H and N) were performed using an Elementar Vario Cario Erballo analyzer. The powder X-ray diffraction (PXRD) measurements were recorded on a Bruker D8 ADVANCE X-ray diffractometer using CuKα radiation (λ = 1.54184 Å) at room temperature from 5° to 50° with a sweeping speed of 10°/min. The single-crystal X-ray data were collected using a Rigaku SuperNova, Dual, Cu at zero, AtlasS2. The structure was solved using the program Olex2 1.3 and refined using a full-matrix least-squares method based on F^2^ using the SHELXL-2018/3 method. Hydrogen atoms were added geometrically and refined using a riding model. The temperature- and field-dependent magnetic susceptibility measurements were carried out using a Quantum Design MPMS XL5 SQUID magnetometer. The IR spectra (KBr tablet) were recorded on the WQF 510A FTIR equipment in the range of 400−4000 cm^−1^ with a sweeping interval of 2 cm^−1^. The UV-Vis absorption spectra were measured using a TU-1901 spectrophotometer in the range of 280−600 nm with a sweeping interval of 0.5 nm. The photoluminescence spectra were measured using a Lengguang F98 fluorescence spectrophotometer. The sweeping speed was 1000 nm/min with a sweeping interval of 1 nm.

## 4. Conclusions

In conclusion, novel high-nuclearity clusters have been obtained using the hydrolysis strategy: saddle-shaped {Gd_7_} and a three-layer double sandwich {Gd_14_}. Both complexes exhibit good magnetocaloric properties with magnetic entropy changes of 17.4 J kg^−1^ K^−1^ for Gd_7_ at 3 K and 5 T and 22.3 J kg^−1^ K^−1^ for Gd_14_ at 2 K and 5 T, respectively. Further work on the new clusters {Ln^III^_7_} and {Ln^III^_14_} (Ln = Dy, Tb, Eu and Ho) are in progress in our laboratory, which possibly behave as SMMs or fluorescent materials. The combination of rare earth ions and new ligands paves the way for the discovery of new polynuclear clusters with unprecedented structures and functionalities.

## Figures and Tables

**Figure 1 molecules-29-00389-f001:**
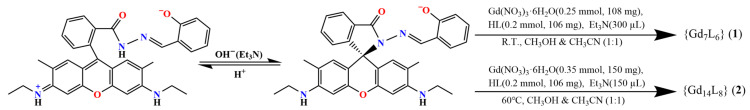
Transformation of ring-opened and ring-closed form of rhodamine-6G-type ligands and their reactions.

**Figure 2 molecules-29-00389-f002:**
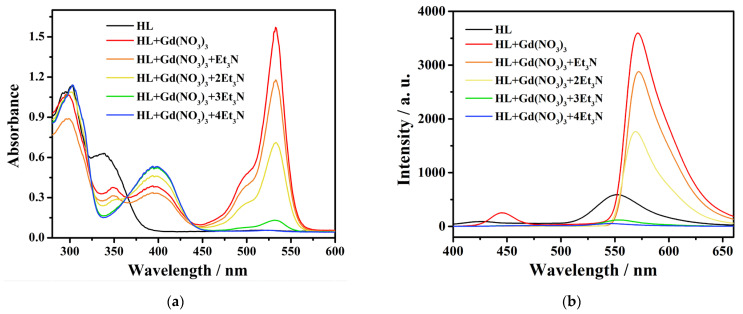
UV-Vis (**a**) and fluorescent spectra (λ_ex_ = 354 nm) (**b**) for the ligand HL and the reaction mixture containing HL and Gd(NO_3_)_3_ with different amounts of Et_3_N in MeOH.

**Figure 3 molecules-29-00389-f003:**
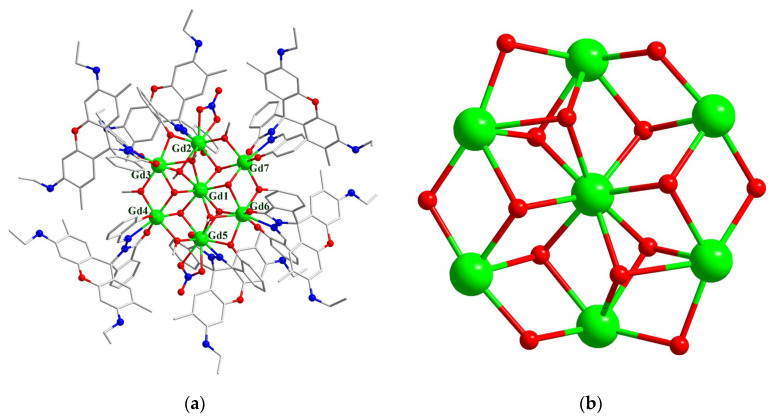
(**a**) The structure of the {Gd^III^_7_} cation for complex **1**. Hydrogen atoms and solvents have been omitted for clarity. (**b**) The core skeleton graph of complex **1**. Color code: Gd^III^ green; O red; N blue; C gray.

**Figure 4 molecules-29-00389-f004:**
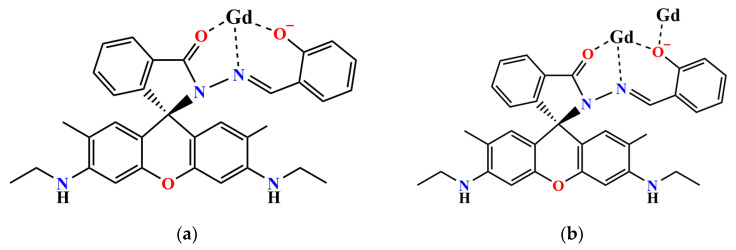
Schematic diagram of two different coordination modes of ring-closed rhodamine ligands L^−^. (**a**) Tridentate chelating mode; (**b**) phenolic oxygen bridging mode.

**Figure 5 molecules-29-00389-f005:**
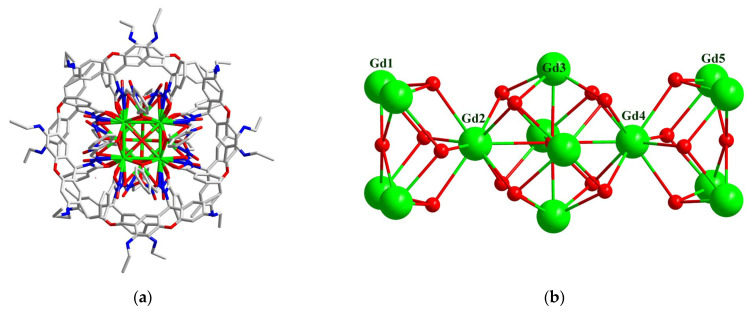
(**a**) The structure of complex **2**. Hydrogen atoms and solvents have been omitted for clarity. (**b**) The core skeleton of complex **2**. Color code: Gd^III^ green; O red; N blue; C gray.

**Figure 6 molecules-29-00389-f006:**
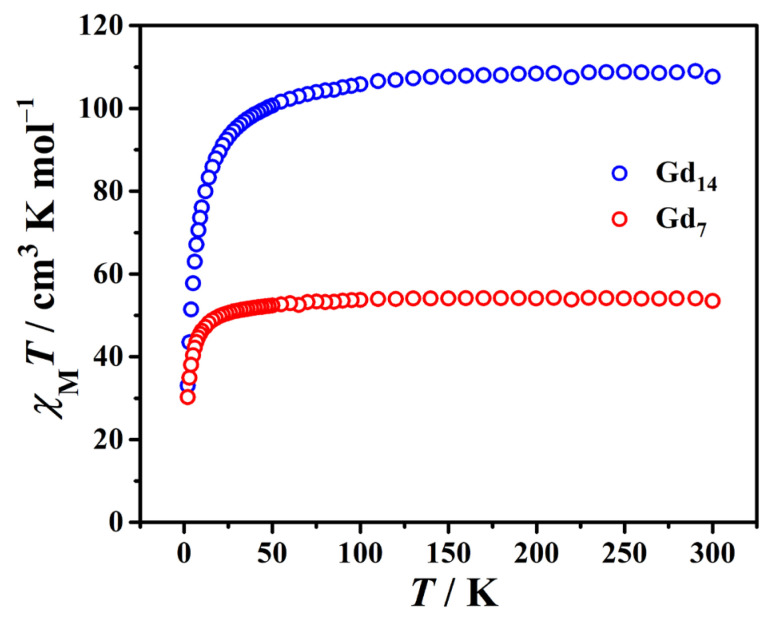
Temperature dependence of *χ*_M_*T* for {Gd_7_} and {Gd_14_} under a 1000 Oe magnetic field in the range of 2–300 K.

**Figure 7 molecules-29-00389-f007:**
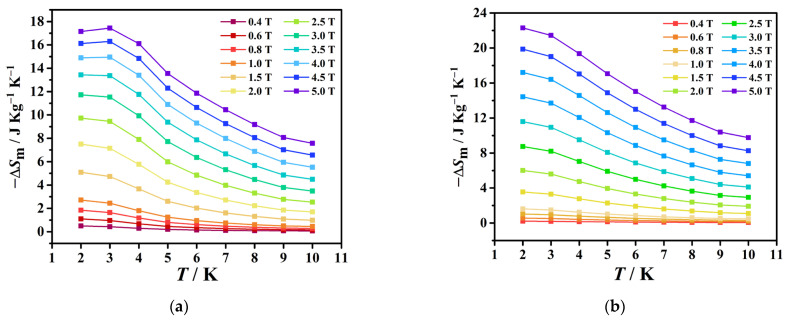
Experimental −Δ*S*_m_ values of **1** (**a**) and **2** (**b**) for multiple temperatures and magnetic fields calculated from magnetization data.

## Data Availability

The data supporting the reported results are available from the corresponding author.

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
