# Peer review of "{GdIII7} and {GdIII14} Cluster Formation Based on a Rhodamine 6G Ligand with a Magnetocaloric Effect"

_molecules, 2024, doi:10.3390/molecules29020389_

Round 1
Reviewer 1 Report
Comments and Suggestions for Authors
The manuscript "{GdIII7} and {GdIII14} clusters based on Rhodamine 6G ligand with magnetocaloric effect" investigated the possibility of novel magnetic complexes incorporating gadolinium and rhodamine 6G ligand.
The research raised an initial solution to improve the magnetic refrigeration effect of gadolinium clusters, which is important in practical applications and relevant in the field. The conclusions were consistent with the evidence and the references appropriate.
There is a question on the spectral changes of rhodamine 6G ligand should be supplemented. As a widely used dye, considering the specific fluorescence properties of rhodamine 6G, is there any spectral changes between rhodamine 6G itself and {GdIII7}/{GdIII14} clusters adopting rhodamine 6G ligand? Also, is it in accordance with the spectral changes of ring-opened and ring-closed structures of rhodamine 6G? Please give a supplement.
Reviewer 2 Report
Comments and Suggestions for Authors
In the manuscript, Kou et al. described the syntheses, crystal structures and magnetic properties of two polynuclear Gd(III) complexes. Gd7 and Gd14 molecules are obtained using the ring-closed rhodamine ligand via the hydrolysis approach. The two complexes display good magnetocaloric properties. The present manuscript is recommended for publication in Molecules as a full paper.
Minor points:
1. In the structure section, the selected bond distances and angles are given in SI, but they are not discussed in the text.
2. Figures S7-S10 are structures for previously reported complexes. The corresponding reference should be given in SI.
3. The calculated Brillouin curve was only given for the data of 2 K. It is necessary to provide the Brillouin curves of 3-10 K.
4. Infrared spectrum data should be provided for characterizing the complexes.
5. The reagent in Figure 1 is Gd(NO3)3·5H2O, but the reagents mentioned in the experimental section is Gd(NO3)3·6H2O. Please check.
Author Response
Reviewer 2
In the manuscript, Kou et al. described the syntheses, crystal structures and magnetic properties of two polynuclear Gd(III) complexes. Gd7 and Gd14 molecules are obtained using the ring-closed rhodamine ligand via the hydrolysis approach. The two complexes display good magnetocaloric properties. The present manuscript is recommended for publication in Molecules as a full paper.
Minor points:
- In the structure section, the selected bond distances and angles are given in SI, but they are not discussed in the text.
Response:Thank you for your helpful suggestions. The discussion on Gd-O bond distances and the bridging Gd-O-Gd bond angles for the two complexes has been added in the text.
- Figures S7-S10 are structures for previously reported complexes. The corresponding reference should be given in SI.
Repsponse:The corresponding reference has been added in the caption of Figures S7-S10.
- The calculated Brillouin curve was only given for the data of 2 K. It is necessary to provide the Brillouin curves of 3-10 K.
Response:The Brillouin curves of 3-10 K have been calculated and added in Supporting Information.
- Infrared spectrum data should be provided for characterizing the complexes.
Response: IR spectra have been measured and deposited as SI. Brief discussion has been made in the text.
- The reagent in Figure 1 is Gd(NO3)3·5H2O, but the reagents mentioned in the experimental section is Gd(NO3)3·6H2O. Please check.
Response: The typo has been corrected in the revised manuscript. Thank you very much.
Reviewer 3 Report
Comments and Suggestions for Authors
Chelating ligand was prepared using rhodamine 6G and salicylaldehyde, and two complexes with different nuclei numbers were constructed, and their stucutural-relationship activity was characterized by single-crystal structure and magnetic tests. The overall research results are complete and innovative, suitable for publication in this journal after minor revision.
1, The synthetic preparation process of the ligand needs to be added.
2, Is the precise guest molecule in the structural formula indicated by single crystal test or thermogravimetric analysis?
3, What is the weak force between different cluster units?
Comments on the Quality of English LanguageMinor editing of English language required
Author Response
Reviewer 3:
Chelating ligand was prepared using rhodamine 6G and salicylaldehyde, and two complexes with different nuclei numbers were constructed, and their stucutural-relationship activity was characterized by single-crystal structure and magnetic tests. The overall research results are complete and innovative, suitable for publication in this journal after minor revision.
1, The synthetic preparation process of the ligand needs to be added.
Response: Thank you for your helpful suggestion. The synthetic process of the ligand HL has been added in the synthesis section of the text.
2, Is the precise guest molecule in the structural formula indicated by single crystal test or thermogravimetric analysis?
Response: Thank you for your helpful suggestions. The Gd7 complex tends to lose solvents, and the TGA data along with the microelemental CHN analyses show the complete removal of solvent. The Gd14 complex is stable, and the TGA data are consistent with the content of the solvents. The TGA data have been deposited in SI.
3, What is the weak force between different cluster units?
Response: Thank you for your helpful suggestions. We have carefully checked the cell packing diagram of the two complexes, and failed to find any apparent intermolecular interaction in the Gd7 complex. However, the Gd14 complex shows hydrogen bonding between the acetonitrile and the HN group of the chelating ligand L-. The discussion on intermolecular coupling has been added in the revised manuscript.